# GeST: Towards Building A Generative Pretrained Transformer for Learning Cellular Spatial Context

## Abstract

Learning the spatial context of cells through pre-training may enable us to systematically decipher tissue organization and cellular interactions in multicellular organisms. Yet, existing models often focus on individual cells, neglecting the intricate spatial dynamics between them. We develop GeST, a deep generative transformer model that is pre-trained on the task of using information from neighboring cells to iteratively generate cellular profiles in spatial contexts. In GeST, we propose a novel serialization strategy to convert spatial data into sequences, a robust cell quantization method to tokenize continuous gene expression profiles, and a specialized attention mechanism in the transformer to enable efficient training. We pre-trained GeST on a large-scale spatial transcriptomics dataset from the mouse brain and demonstrated its performance in unseen cell generation. Our results also show that the pre-trained model can extract spatial niche embeddings in a zero-shot way and can be further fine-tuned for spatial annotation tasks. Furthermore, GeST can simulate gene expression changes in response to spatial perturbations, closely matching experimental results. Overall, GeST offers a powerful framework for generative pre-training on spatial transcriptomics.

## 1 Introduction

In recent years, pre-training transformer-based models on large-scale scientific data have emerged as a new paradigm in AI for biology (Webb et al., 2018; Bunne et al., 2024; Szałata et al., 2024), enabling the development of foundation models tailored to specific modalities such as DNA sequences (Nguyen et al., 2024), proteins (Abramson et al., 2024), and single-cell gene expression (Theodoris et al., 2023; Hao et al., 2024; Cui et al., 2024; Bian et al., 2024). However, most of these models focus on gene-gene relationships or products within isolated cellular contexts, neglecting the intricate cell communications in spatial that is fundamental in multicellular organisms. As a result, current models struggle to handle spatial tasks or understand spatial patterns, which limits their ability to fully comprehend and model cellular behaviors in complex tissue environments.

Spatial transcriptomic (ST) is an emerging technology that combines high-throughput gene expression profiling with spatial localization of cells within tissue sections (Moses & Pachter, 2022). Beyond scRNA-seq data, where a cell is analogous to a sentence composed of gene tokens, in spatial transcriptomics data, a tissue is a document consisting of many cell sentences. Rich ST datasets enable us to learn cell-cell relationships in a data-driven manner. Previous studies such as GraphST(Long et al., 2023) and SpaGCN (Hu et al., 2021) often trained graph neural network to integrate spatial and gene expression information. These models were trained independently for each dataset, leaving the paradigms of pretraining or generative modeling unexplored. A recent study called CellPLM (Wen et al., 2023) built a BERT-style (Devlin, 2018) pre-trained model by using partial gene expression data from a target cell and information from its neighboring cells to predict the remaining gene expression. However, since CellPLM needs to know the expression of a subset of genes in a cell before predicting the cell's overall gene expression, it cannot generate brand new cells in unseen locations. This limitation restricts its ability to explicitly study how spatial context alone influences a cell's characteristics, which is crucial to understand the pattern of tissue functionality. In addition, constrained by the BERT modeling, its predictions are based on the existing input all at once, lacking the ability to iteratively generate new cells or adapt to dynamic spatial contexts.

Inspired by the advancements of GPT models (Achiam et al., 2023; Radford et al., 2019; Brown, 2020), we endeavor to develop a generative pre-trained model on ST data to overcome these limitations. Such a model can iteratively generate cells at unseen positions. It can further investigate perturbation effects in spatial contexts by manipulating the given neighborhood information, providing an *in-silico* extension of current single-cell perturbation studies. However, GPT modeling on ST data faces several unique challenges. First, there is no inherent order of cells within two-dimensional tissue sections. While one solution can involve serializing spatial data into a fixed sequence, this approach fails to accommodate scenarios requiring different orders during inference. Therefore, a flexible serialization strategy is essential. Second, cells in spatial transcriptomics data have continuous gene expression profiles. Unlike the discrete tokens in natural language, these continuous values may introduce error accumulation during the autoregressive generation (Figure A.1).

To address these challenges and support new applications such as *in-silico* spatial perturbation, we present GeST, a deep generative pre-trained transformer that iteratively generates cells by leveraging the neighbor information. To the best of our knowledge, GeST is the first generative pre-trained transformer to understand cell-cell relationships and advance cell modeling in spatial context. Our experiments showed its superior performance across several downstream tasks. Our work makes the following key contributions:

- **Spatial Serialization Strategy**: We introduce a novel method for serializing spatial transcriptomics data, coupled with a specialized attention mechanism called *Spatial Attention* and a designed input sequence for the transformer. This ensures high computational efficiency during pre-training and provides flexibility during inference.

- **Robust Cell Tokenization**: We develop a cell quantization method to tokenize cells' expression profiles, alongside a hierarchical pre-training loss designed to mitigate error accumulation in autoregressive generation.

- **Transferable Performance**: We pre-train a GeST model with 1 million parameters and demonstrated that the pre-trained model can achieve superior performance after being transferred to clustering and annotation tasks.

- **Pioneering Spatial Perturbation Analysis**: We establish GeST as a pioneering model for *in-silico* spatial perturbation analysis, achieving substantial alignment with results from real spatial experiments.

## 2 TASK FORMULATION

Given a spatial omics dataset, we denote it as a set $\{x_1, x_2, x_3, \ldots, x_n\}$, encompassing all $n$ cells within a two-dimensional tissue slice. We define two critical functions: $g(\cdot)$, the gene expression retrieval function, and $s(\cdot)$, the spatial information retrieval function. For any given cell $x$, the set $N(x) = \{x_{N1}, x_{N2}, \ldots, x_{Nk}\}$ includes all $k$ neighboring cells.

**Unseen cell generation (Pre-training task)**. The objective is predicting the gene expression $g(x)$ of a target cell $x$ based on its spatial location. Instead of making a direct prediction based on the spatial coordinate like $P(g(x)|s(x))$, we aim to predict the gene expression of a target cell $x$ using the spatial locations and gene expressions of its neighboring cells:

$$P(g(x)|s(x), g(N(x)), s(N(x))) \tag{1}$$

Following this modeling, the objective function of our task is:

$$\min_\theta ||g(x_{k+1}) - \mathcal{F}_\theta(x \mid s(x), g(N(x)), s(N(x)))|| \tag{2}$$

where $\mathcal{F}_\theta$ represents our proposed spatial generative model. However, spatial data lacks a natural sequential order, which challenges the application of auto-regressive models that usually work for sequence prediction tasks. To address this, we transform this objective into a sequential format:

$$\min_\theta ||g(x_{k+1}) - \mathcal{F}_\theta(x_{k+1} \mid s(x_{k+1}), g(x_1), s(x_1), g(x_2), s(x_2), \ldots, g(x_k), s(x_k))|| \tag{3}$$

where $x_{k+1}$ is the target cell, and the sequence $(x_1, x_2, \ldots, x_k)$ represents its neighbors, arranged by a serialization strategy.

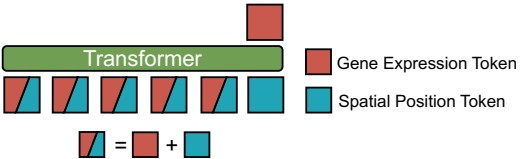

Figure 1: Given one spatial location $x$, the spatial generation model takes its spatial coordinate information $s(x)$, its neighbors' $N(x)$ gene expression $g(N(x))$ and spatial location $s(N(x))$ as the input, and predicts its gene expression value.

Furthermore, similar to natural language generation, the objective of this task can be extended to generating multiple cells, which requires iteratively applying the function $\mathcal{F}$ in Equation 3 to progressively estimate the gene expression of cells adjacent to the known tissue boundaries.

**Niche clustering/annotation**. In spatial transcriptomics, a niche refers to a functional or structural tissue region where cells interact with each other and their surroundings. Identifying and understanding these niches is crucial for elucidating tissue organization (Jain & Eadon, 2024). Unlike the spatial generation, the objective of niche clustering or annotation task is to map the spatial and gene expression information of cell $x$ and its neighbors $N(x)$ information into a high-dimensional embedding space that facilitates clustering or label prediction. This encoding process can be formalized as follows:

$$E_\phi(x, N(x)) = \mathcal{F}_\phi(g(x), s(x), g(N(x)), s(N(x))) \tag{4}$$

Here, $E_\phi$ represents the encoding function parameterized by $\phi$, which integrates the gene expression and spatial information of a cell and its neighbors into a unified embedding vector.

***In-silico* spatial perturbation**. This task aims to simulate gene expression changes in response to the perturbation of given target cells in the spatial context. We maintain the spatial positions of the target cells and their neighboring cells but manipulate the gene expressions of the target cells to predict how the neighboring cells change using Equation 3. Since it is impossible in real-world experiments to obtain both normal and perturbed gene expressions from the same cell simultaneously, we assess our results by analyzing statistical variations in gene expression between the normal and perturbed scenarios and corroborate these findings with knowledge from existing literature.

## 3 GENERATIVE PRE-TRAINING AND FINE-TUNING METHODOLOGY

We introduce GeST, a spatial cell language model, with the following basic components: a cell expression quantization module and a transformer decoder. During pre-training, only the neighboring cells have complete information (both gene expression and spatial position tokens), while the target cells are provided only with their spatial position token, compelling the model to predict the gene expression at that specific location(Figure 1). Fine-tuning extends GeST to other downstream tasks with niche embeddings. We introduced these in detail in the following sections.

### 3.1 SERIALIZATION STRATEGY

At each training step, we first crop a square from the training tissue section, and all $n$ cells $\mathcal{X} = \{x_{o1}, x_{o2}, x_{o1}, ..., x_{oN}\}$ in this square will be used to constitute a training sequence. We serialize them by sampling along diagonal paths. Specifically, we first randomly select an anchor point $\boldsymbol{p}$ from four vertexes of the square. Then we calculate all cells' Euclidean distances from $\boldsymbol{p}$ and use them as the sampling weights: $\{w_{o1}, w_{o2}, w_{o3}, ..., w_{oN}\}$, where $w_{oi} = ||s(x_{oi}) - \boldsymbol{p}||_2$. We do sampling without replacement in $N$ times and thus get a sequence $[x_1, x_2, x_3, ...x_N]$. At each sampling time $t$, the probability of selecting cell $x_{oi}$ is:

$$P(x_t = x_{oi}) = \frac{w_{oi}}{\sum_{j \in \mathcal{X} \backslash \mathcal{D}} w_{oj}} \tag{5}$$

where $\mathcal{D}$ contains cells that have been selected into the sequence. This strategy allows the adjacent cells in spatial to have similar indexes in sequence but still retain randomness to prevent the model's overfitting. Based on this, we design a novel attention mechanism to enhance the computational efficiency of the pre-training, as shown in section 3.3.

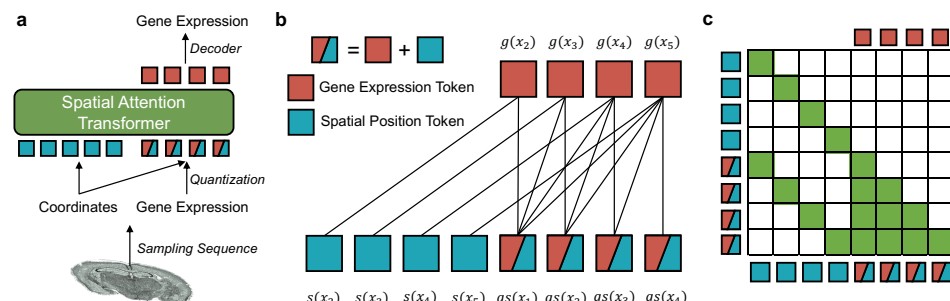

Figure 2: Model architecture. a) Schematic overview of GeST. b) illustration of output and input relationships in our pre-training task. c) Spatial Attention matrix.

## 3.2 CELL TOKENIZATION

Given a training sequence, we tokenize both gene expression and spatial position for each cell. For the spatial position, we take the cell that is centered in the original tissue region as the origin of the coordinate system. Then we normalized each cell's coordinates by calculating the relative coordinates to the origin. We tokenize the coordinate values by a two-dimensional sinusoidal positional encoding (Detailed in A.3).

For gene expression, we found in preliminary experiments that directly generating continuous single-cell expressions would cause error accumulation in the iterative generation process, eventually leading to model failure (refer to ablation study "w/o quantization" in Table 4). Therefore, we propose to build a "meta cell vocabulary" to quantize cells' continuous expression to discrete cell states. Formally, given a training spatial dataset $\mathcal{X} = \{x_1, x_2, x_3, \ldots, x_n\}$ with $n$ cells and $T$ genes, we first perform PCA reduction to $p$ dimensions and categorize them into $K$ clusters by K-means. The center point of each cluster contributes a "meta cell", and there are two attributes of the meta cell vocabulary: the mean expression $\mathcal{C}_{\mathrm{expr}} \in \mathbb{R}^{K \times T}$ and the mean PCs $\mathcal{C}_{\mathrm{pca}} \in \mathbb{R}^{K \times p}$ (Algorithm 1). We note that this quantization of continuous value loses distance relationship in the expression space, i.e., two different meta cell labels may stand for either two very similar or two totally different meta cells. Thus, we further perform K-means clustering on the meta cell vocabulary with few cluster numbers to obtain hierarchical labels $L_1, L_2, L_3$ at various levels. After that, for any input continuous single-cell expression $\boldsymbol{y} \in \mathbb{R}^T$, we can project its expression vector to PCA space and retrieve the nearest meta cell. Then, we substitute the original expression with its corresponding meta cell's mean expression $\boldsymbol{c} \in \mathbb{R}^T$ as the actual input to the model (Algorithm 2).

## 3.3 SPATIAL CONTEXT-AWARE DECODER

Our main model is a transformer decoder (Vaswani, 2017) modified for the spatial generation task. During pre-training, the model's input is divided into two contiguous sequences after tokenization: the neighbor cell sequence and the target cell position sequence, constituting a sequence of $N + (N - 1)$ tokens. The output is a sequence of $N - 1$ gene expression (Figure 2a).

**Neighbor Cell Sequence**. This part consists of the complete tokens of the first to the $(N - 1)$-th cells, totaling $N - 1$ cells. Each token combines both gene expression and spatial information: $[gs(x_1), gs(x_2), \ldots, gs(x_{N-1})]$, where $gs(x_i) = g(x_i) + s(x_i)$ represents the combined embedding of gene expression and spatial position for cell $x_i$.

**Target Cell Position Sequence**. This part has the spatial position tokens of the second to the $N$-th cells, totaling $N - 1$ cells, formulating a target cell position token sequence, $[s(x_2), s(x_3), \ldots, s(x_N)]$.

As illustrated in Figure 2b, for each target cell position $s(x_{i+1})$, we use the transformer decoder's parallel training capability to predict the gene expression of the next neighbor cell $g(x_{i+1})$. Prediction of target cell $x_{i+1}$ is conditioned on the complete tokens of the neighbor cells $\{gs(x_1), gs(x_2), \ldots, gs(x_i)\}$ and the spatial position token $s(x_{i+1})$.

To achieve this, we design a special attention matrix called ***Spatial Attention***. Unlike the causal attention used in language models, which employs a lower triangular mask to ensure that each token can only attend to previous tokens in the sequence, our Spatial Attention allows each position to attend to specific relevant tokens, enabling the model to capture spatial dependencies more effectively (Figure 2c). Specifically, for a sequence length of $2L$ (where $L = N - 1$), the attention mask $M$ is a $2L \times 2L$ matrix. For the token at position $i + L$ (corresponding to predicting the gene expression of cell $x_{i+1}$), we allow attention to 1) The neighbor cell tokens at positions 1 to $i$ (i.e., $\{gs(x_1), gs(x_2), \ldots, gs(x_i)\}$). and 2) The target cell position token at position $i + L$ (i.e., $s(x_{i+1})$). Formally, for $i \in [1, L]$, the attention mask $M$ is defined as:

$$M_{i+L,t} = \begin{cases} 1, & \text{if } t \in \{1, 2, \ldots, i\} \cup \{L + i\} \\ 0, & \text{otherwise} \end{cases} \tag{6}$$

This design leverages the transformer decoder's capability for parallel computation while effectively modeling spatial relationships (Radford et al., 2019). By allowing each prediction to attend to the relevant neighbor cells and the spatial position of the target cell, the model learns to generate gene expressions conditioned on spatial context. After the decoder, a multilayer perceptron is used to convert the hidden embedding $\boldsymbol{h} \in \mathbb{R}^D$ to gene expression space $\hat{\boldsymbol{y}} \in \mathbb{R}^T$. Each element of prediction $\hat{\boldsymbol{y}}$ represents the expression level of a specific gene, where $T$ is the total number of genes.

### 3.4 Loss function

To compute the loss between the predicted $\hat{\boldsymbol{y}}$ and the ground truth gene expression $g(x)$, instead of computing the regression loss, we propose a hierarchical cross-entropy loss function. From the previous section, we quantize continuous gene expression vectors into discrete categories from a meta cell vocabulary $\mathcal{C}$. Each meta cell $c \in \mathcal{C}$ is associated with hierarchical labels at four levels: $l_0(c)$, $l_1(c)$, $l_2(c)$, and $l_3(c)$, each of them has $K$, $K_1$, $K_2$, and $K_3$ categories. For $K_i$, we use 15, 10, and 5 as default. Then we project the model outputs $\hat{\boldsymbol{y}}$ to logits $\mathbf{z} \in \mathbb{R}^K$ corresponding to each meta cell:

$$\mathbf{z} = \hat{\mathbf{y}}\mathbf{W}^\top \tag{7}$$

where $\mathbf{W} \in \mathbb{R}^{K \times T}$ is the codebook matrix containing the meta cell embeddings. We apply the softmax function to obtain the predicted probability distribution over the meta cells:

$$p(c) = \frac{\exp(z_c)}{\sum_{c' \in \mathcal{C}} \exp(z_{c'})} \tag{8}$$

To compute the hierarchical losses, we aggregate the probabilities over meta cells to obtain probabilities over hierarchical labels at each level. For hierarchical level $i$, the probability of category $k$ is calculated as:

$$p^{(i)}(k) = \sum_{c \in \mathcal{C}} \delta\left(l_i(c) = k\right) p(c) \tag{9}$$

where $\delta(\cdot)$ is the Kronecker delta function, which equals 1 if the condition is true and 0 otherwise.

The overall loss function $\mathcal{L}$ is defined as a weighted sum of the negative log-likelihood losses at each hierarchical level:

$$\mathcal{L} = \sum_{i=0}^{3} \alpha_i \cdot \mathcal{L}_i = \sum_{i=0}^{3} \alpha_i \cdot \left(-l_i(y) \log p^{(i)}\right) \tag{10}$$

where $\alpha_i$ are weights and we set 0.25 as default. $\mathcal{L}_i$ is the cross-entropy loss at level $i$, and $l_i(y)$ is the ground truth hierarchical label of the target cell's meta cell at level $i$. We minimize the loss function $\mathcal{L}$ across all training samples during training. Under this hierarchical loss function, the model is encouraged to make correct predictions at multiple levels, making it more robust to wrong predictions in a single layer, especially on the finest layer.

With regard to the inference strategy, there are two modes to convert the predicted probability to the final predicted gene expression value: (1) "picking" mode: we directly use the meta cell's expression with the highest probability as the prediction. (2) "weighted aggregation" mode: we set $p(c)$ as the weight to aggregate all meta cells' expression as the prediction.

Figure 3: Niche embedding. a) We input all cells' information and extract the last token as niche embedding. b) for the fine-tuning task, a pooling operation is used for token aggregation.

### 3.5 Niche embedding extraction

GeST can be used for niche clustering in a zero-shot manner and can be fine-tuned for niche annotation, both of which require the extraction of a niche embedding. Given a niche comprising a target cell $x$ and its $N-1$ neighboring cells, we follow the pre-training setup to generate both position tokens and cell tokens as input. We include the target cell's position token twice: once at the beginning and once at the end of all position tokens, resulting in a total of $N+1$ position tokens. For the cell sequence, we incorporate the content tokens of all cells, yielding $N$ tokens, as illustrated in Figure 3a. Under this configuration, the last output token of the model is generated based on the information from all cells, and we define this as the niche embedding, consistent with Equation 4. We use the hidden embedding $h$ obtained here for the zero-shot niche clustering task.

For fine-tuning, we input the sequence in the same setting and apply a pooling operation to aggregate these embeddings into a single vector (Figure 3b) $h_p = \text{Pool}(\{h_x\} \cup \{h_n \mid n \in N(x)\})$. Mean pooling is default but we also compare it with max pooling in downstream tasks. The embedding $h_p$ is then passed through a linear layer to obtain the logit for the niche classification task.

## 4 Experiments

We conducted four experiments to demonstrate our model's capabilities: unseen region gene expression generation, zero-shot niche embedding clustering, fine-tuning based niche label annotation, and *in-silico* spatial perturbation. In generations, we validated our model on three different datasets: a large-scale MERFISH spatial dataset of the whole mouse brain (Zhang et al., 2023), a Visium human primary liver cancer (PLC) dataset (Wu et al., 2021), and a stereo-seq brain dataset (Cheng et al., 2022). The MERFISH dataset contains more than 3 million cells and has two replicates, and we used the first Mouse1 and second Mouse2 replicate as the training and test set, respectively. The rest two PLC dataset and the Stereo-seq dataset are used to show GeST ability to handle irregular spatial patterns and various resolutions. For the *in-silico* spatial perturbation experiment, we applied our pre-trained model to simulate the mouse brain ischemic process. We used a real ischemic spatial dataset (Han et al., 2024) as a reference, providing a basis for ischemia-induced genes as ground truth. Due to differences in resolution and gene panel between our training and the reference datasets, instead of applying our model to this dataset, we performed *in-situ in-silico* gene perturbation on a section of our training dataset that closely matched the spatial location of this ischemic dataset. This allowed us to identify proposed ischemia-induced genes and compare our findings with established results. More details can be found in Section 4.4.

We implemented GeST with 8 Transformer layers and 8 heads per layer, having about 1 million parameters. Each training sequence comes from a sampled square region with a side length of 600μm. We used PyTorch (Paszke et al., 2019) and trained the model on four NVIDIA A800 GPUs. The pre-training process took approximately 3 hours.

### 4.1 Unseen cell generation

For pre-training, we selected one right hemisphere section from the Mouse1 dataset as validation data and used the remaining 146 Mouse1 left hemisphere sections as the training data, totaling

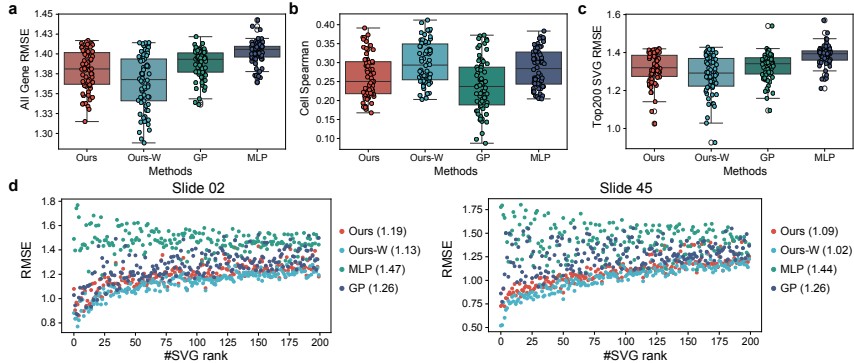

Figure 4: Gene expression prediction results on unseen region. a) Root Mean Square Error(RMSE) of all genes. Each dot represents the mean error of all genes calculated from a slide for each method. The bar plot shows the errors across all slides. b) Spearman correlation of all genes within cell. Each dot and the barplot is defined in a similar way. c) RMSE of top 200 spatial variable genes. d) Each dot represents a gene; the x-axis is the rank of spatial variation, and the y-axis is the RMSE.

Table 1: AMI score of different methods niche clustering results at both the region and division level. NicheC: NicheCompass, Ours-FT: Our finetuned model

| Level | Ours | GraphST | NicheC. | SpaGCN | STAGATE | Raw | Ours-FT |
|---|---|---|---|---|---|---|---|
| Division | **0.469** ±0.173 | 0.388 ±0.152 | 0.438 ±0.177 | 0.201 ±0.070 | 0.420 ±0.167 | 0.183 ±0.091 | **0.470** ±0.174 |
| Region | **0.484** ±0.107 | 0.414 ±0.091 | 0.481 ±0.113 | 0.231 ±0.067 | 0.462 ±0.114 | 0.244 ±0.077 | **0.515** ±0.077 |

2,839,984 cells (see description of datasets in A.1). For testing, we simulated unseen regions on each coronal section from the Mouse2 dataset by randomly cropping squares with side lengths ranging from 300μm to 900μm. We recorded the spatial coordinates of the cropped cells as model inputs, enabling direct comparison between the predicted and actual gene expressions. If the unseen region size exceeded the maximum neighbor size used for model training, we iteratively generated cells (see A.2 for more details). We trained two models as baselines, a Gaussian Process and a Multi-layer Perceptron (MLP), by using cells' absolute spatial coordinates and gene expressions from the uncropped areas in each slide as training data.

As shown in Figure 4a, our model in the "picking" mode (labeled as "Ours"), which directly retrieves the meta-cell expressions for predictions, exhibited lower regression errors compared to the baseline models. Switching to the "weighted aggregation" mode (labeled as "Ours-W"), our model produced more variable outputs and achieved even lower regression errors. Spearman correlation analysis (Figure 4b) further confirmed that "Ours-W" achieved the highest performance in predicting ground truth gene expressions within cells. Recognizing that not all genes exhibit strong spatial patterns, we focused on the top 200 spatially variable genes (SVGs) per slide, identified using SOMDE (Hao et al., 2021). Our model consistently achieved the lowest prediction errors on these SVGs (Figure 4c). In a detailed analysis of slides 2 and 45 (Figure 4d), our model more accurately predicted genes with high spatial variation, a trend less evident in the baseline models. We also visually compared the predicted and actual spatial patterns of SVGs (Figure A.2) and noted that our model could predict well-aligned patterns. These findings highlight our model's capability of learning the underlying spatial characteristics of gene expression and cell organization.

For the 10X Visium PLC dataset, on the evaluation slide we cropped an area containing the edge of the tumor as unseen spots (Figure A.3). In the Stereo-seq brain dataset, we cropped an area containing all the brain cortex layers (see A.1 and Figure A.4 for details). Compared with MLP and GP on all these two datasets, our model achieves the highest performance. Specifically, in PLC, the marker genes of malignant cells (*SPINK1, GPC3, AKR1B10*) and fibroblasts (*COL1A1, COL1A2*) are predicted to have clear zones, which are consistent with the ground truth (Figure A.5). These results reveal GeST generalizable generation ability on data from various techniques.

Table 2: F1 score of annotation results. We report the mean ± standard deviation.

| Level | Ours (mean) | Ours (max) | scANVI | CellTypist |
|---|---|---|---|---|
| Division | **0.585±0.153** | 0.579±0.150 | 0.316±0.072 | 0.162±0.060 |
| Region | **0.407±0.902** | 0.396±0.084 | 0.202±0.058 | 0.051±0.020 |

## 4.2 UNSUPERVISED NICHE CLUSTERING

A key feature of our model is its ability to learn the spatial context of neighboring cells during pre-training. We evaluated this capability on spatial clustering and annotation tasks. We used two levels of anatomical labels, "Division" and "Region", provided by the Mouse Brain Common Coordinate Framework (CCF) v3 (Wang et al., 2020), as ground truth. For the niche clustering task, we used niche-level embeddings of each cell for clustering. We compared our method with three methods: NicheCompass (Birk et al., 2024), STAGATE (Dong & Zhang, 2022), GraphST (Long et al., 2023) and SpaGCN(Hu et al., 2021). We also included a baseline that gets clusters based solely on the cell's own gene expression data (Raw). All spatial clustering methods outperformed the raw baseline, with our model achieving the highest adjusted mutual information (AMI) scores at both resolutions (Table 1 and Figure A.6). These results demonstrate that our pre-trained model can be effectively transferred to new tissues in a zero-shot manner. We also noted that after we continued training our model on the test data in the same generative way, the model showed an even higher performance.

Unlike previous methods, our model allows control over the scope of niche information used to generate niche embeddings. To investigate this, we used the same pre-trained model but varied the neighborhood window size during inference to 200μm, 400μm, and 600μm. As shown in Table A.2, increasing the window size improved clustering performance, a trend observed in both zero-shot and fine-tuning scenarios. These findings indicate that our pre-trained model is an effective niche embedding extractor and benefits from incorporating larger neighborhood information.

## 4.3 SUPERVISED NICHE ANNOTATION

We used division and region labels from the Mouse1 dataset as ground truth to fine-tune our model for predicting these labels in the Mouse2 dataset. Before the release of the data we used, spatial annotation methods were lacking due to limited data. So we compared two single cell annotation methods: scANVI (Xu et al., 2021) and Celltypist (Domínguez Conde et al., 2022), and experimented with GeST using both max and mean pooling strategies. As detailed in Table 2, our model outperformed the single-cell methods across both pooling strategies, with the mean strategy achieving a higher performance. Visualizations of the annotation results on a representative section (Figure A.7) show that our model provides consistent annotations across adjacent spatial regions and delineates clear boundaries between different regions.

## 4.4 IN-SILICO SPATIAL PERTURBATION

GeST is a pioneer model for predicting cell response of *in-silico* perturbation in spatial transcriptomic. Inspired by single-cell large models (Theodoris et al., 2023), we proposed a design for *in-silico* spatial perturbations: Firstly, select a region of interest (ROI) and generate the surrounding cell expression as an *in-silico* control group. Next, simulate the perturbation by modifying specific gene expression of cells in that ROI, and predict the expression profiles at the same surrounding positions to obtain an *in-silico* perturbation group (Figure 5a). By analyzing these simulation data, we could study the impact of the perturbations in spatial context.

Here, we demonstrated an *in-silico* spatial perturbation experiment of an ischemic condition of the mouse brain. Recently, Han et al. (2024) measured gene expression and cell distribution in the ischemic mouse brain and identified several ischemic regions including the infarct core area (ICA) and the proximal region of the peri-infarct area (PIA_P). In our experiment, we selected an area with a similar location of ICA from sample 49 from Mouse1 as ROI and generated the surrounding cell expression as the control group, representing the normal gene expression around the ROI. In order to simulate the ischemic effect, we manually altered gene expression in this ROI according to the differentially expressed genes (DEGs) of ICA through ***in-silico* activation** and ***in-silico* inhibition**

(Detailed in A.5). We then fed the model with the same surrounding positions and obtained the gene expression prediction as the perturbation group, representing the predicted PIA_P.

To validate the results, we calculated the Pearson correlation coefficients (PCCs) between gene expression of the perturbation group and the average expression of PIA_P, and compared them with those between control group and PIA_P. The PCCs are significantly higher in the perturbation group (Figure 5b). Taking all 87 high and low DEGs in PIA_P as ground truth, we correctly classified 70.11% of them by our *in-silico* perturbation experiment (Detailed in A.3). This is higher than the baseline accuracy of 44.8%, which is obtained by simply adopting the DEGs from ICA (i.e. a naive model that believes changes in ROI are the same in the neighbor). For example, *Rnh1*, a gene for normal homeostasis of the brain (Hedberg-Oldfors et al., 2023), and *Neurod6*, a key gene in the development and function of the central nervous system (Tutukova et al., 2021), were not modified in ROI, but we predicted them as the high/low DEG in the neighbor area (Figure 5c,d). These findings are consistent with Han et al. (2024). Taken together, GeST demonstrates the ability to simulate perturbation in spatial transcriptomics.

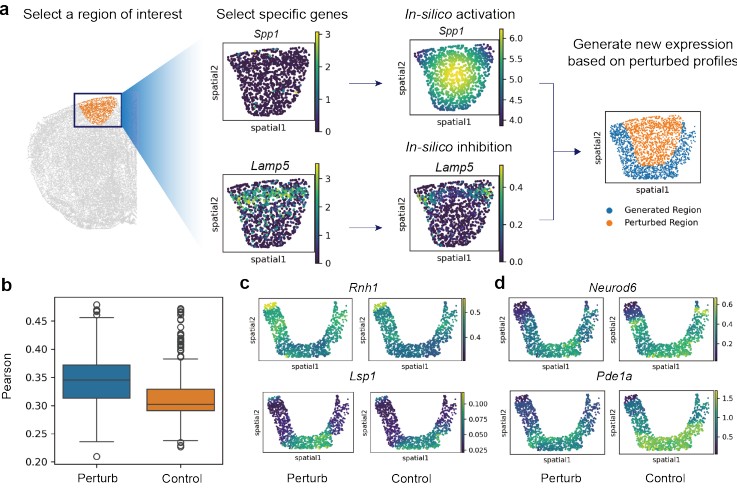

Figure 5: *In-silico* spatial perturbation experiment. a) Flow chart of *in-silico* spatial perturbation experiment. b) Box plot showing Pearson correlation coefficient between perturbation group and control group with PIA_P. *P*-value $< 0.001$, t-test. c) Visualization of highly expressed genes in the perturbation group. d) Visualization of lowly expressed genes in the perturbation group.

## 5 ABLATION STUDY

We conducted ablation experiments to evaluate the impact of model size and training data volume on performance. All models were trained on the left hemisphere sections of Mouse1, with the right hemisphere section split into validation and test sets. Performance was assessed using RMSE for all genes, RMSE for the top 50 spatially variable genes (SVGs), and cell-level Spearman correlation on the test set. We first ablated the model size (Table 3) and observed that increasing the model from a small one to our baseline resulted in significant performance improvements. Beyond our baseline, further increases in model size yielded diminishing benefits, suggesting that our current model strikes an optimal balance between performance and computational efficiency. In addition, we explored the impact of training data size by training models on uniformly sampled subsets comprising half and one-third of the full training data. The results revealed that models trained on larger datasets performed better, suggesting that increasing the amount of training data could further enhance model effectiveness in future work.

Neighbor window size is another important factor that controls the information density of the input and also affect the input's sequence length. We varied this setting from 200μm to 800μm and the corresponding average sequence length was changed from 50 to 1200. In general we found that the model with window sizes of 600μm and 800μm achieved higher performance than that of 200μm (Table 4), demonstrating a large window size allows the model to better learn the underlying spatial

Table 3: Ablation study of model and data size scaling law.

|  | Ours (L8H8&alldata) | Model Size | | | Data | |
|---|---|---|---|---|---|---|
|  |  | L2H2 | L4H4 | L16H16 | 1/2 | 1/3 |
| RMSE | 1.367 | 1.371 | 1.373 | **1.362** | 1.381 | 1.375 |
| RMSE50 | 1.214 | 1.243 | 1.249 | **1.208** | 1.261 | 1.246 |
| Spearman | 0.29 | 0.288 | 0.289 | **0.291** | 0.289 | 0.288 |

Table 4: Ablation study of training window size, loss function, quantization and serialization strategy.'800μm+L' is a model with 12 layers and 8 heads trained on 800μm window size.

|  | Ours | Window size | | | w/o | w/o | random |
|---|---|---|---|---|---|---|---|
|  | (600μm) | 200μm | 800μm | 800μm+L | hierarchy | quantization | order |
| RMSE | 1.367 | 1.376 | 1.371 | **1.362** | 1.382 | 1.389 | 1.384 |
| RMSE50 | 1.214 | 1.251 | 1.232 | **1.204** | 1.27 | 1.315 | 1.256 |
| Spearman | 0.29 | 0.275 | 0.285 | **0.292** | 0.288 | \ | 0.287 |

context of cells. We also noted that the 800μm window size didn't introduce a higher performance, so we trained a model with larger size (12 layers and 8 heads) and found that it achieved a highest performance. These results illustrated the a larger window size may introduce too much spatial variation, making the small model hard to learn. And our current model and data setting is a trade off between them. This result demonstrated the effectiveness of the quantization module on simplifying and regularizing the data space, making the gain in performance.

We also ablated the hierarchy loss and expression quantization module. As shown in Table 4, We found that the model without hierarchy loss achieved a bad performance on all three metrics. For quantization module ablation, we trained a model with the mean square error(MSE) loss. We noted this MSE model generated invalid negative expression value in prediction and showed a quite low performance. We tried to clip the value into a positive number but found that after clipping it predicted all zeros vectors for some cells, causing the failure in computing Spearman correlation. Finally, we compared our spatial ordinal serialization strategy with a random sampling strategy (Figure A.8) and observed a substantial improvement from our strategy. It was intuitively aligned with our assumption that the ordinal strategy meets the actual testing scenario and thus brings the gain in the performance. We also removed all neighbor information by replacing all positional embedding with all-ones vector (Table A.4). This resulted in a significant performance drop, emphasizing the critical role of spatial positional embeddings in enhancing the model's spatial understanding.

## 6 CONCLUSION

Understanding the spatial context of cells is critical for deciphering tissue organization mechanisms (Palla et al., 2022) and has the potential to facilitate the identification of therapeutic biomarkers (Zhang et al., 2022). In this work, we introduced GeST, a novel deep generative pre-trained transformer model that leverages spatial neighbor information, marking the first generative model in the spatial transcriptomics field. GeST employs a cell quantization module to overcome the challenge of error accumulation during generation and utilizes an ordinal serialization strategy with an efficient attention mask design to model two-dimensional data in a sequential generative pretraining framework. Our results demonstrate that this generative task enables the model to learn underlying spatial contexts, thereby enhancing performance on niche-level tasks. To the best of our knowledge, GeST is the first data-driven model to explore perturbation effects in spatial transcriptomic, laying the groundwork for building more comprehensive foundation models for spatial biology.

Despite its advancements, GeST has certain limitations that warrant further investigation. The shortage of large-scale reference ST data may restrict the model's ability generalizability across diverse tissue types. Besides, our current design does not fully account for dynamic gene-gene interactions which similfies the biological mechanism. Future work could integrate GeST with single-cell foundation models to capture both intrinsic and extrinsic cellular characteristics.

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

## A APPENDIX

### A.1 DATASET AND TASK DESCRIPTION

The original MERFISH spatial transcriptomics dataset has two coronal and two sagittal sliced adult mouse brain replicates with 1122 genes, containing 2.8, 1.2, 1.6 and 0.16 million cells, respectively. All these data are mapped to the whole mouse brain taxonomy and Allen CCFv3 and each cell has a 3-dimensional coordinate. The x and y coordinates are experimentally measured and aligned to CCF, and the z coordinates are estimated. Then the multi-tissue annotation can be obtained based on the locations. Since the spatial coordinates value is an important guidance to our model, we only used the measured x and y coordinates in our training. And we selected the first coronal replicate which has the most number of cells as the training data, and used another one as the test data. The first Mouse1 dataset encompasses 2.8 million cells across 147 coronal sections, each annotated with a panel of 1122 genes. The Mouse2 dataset is from a separate mouse brain replicate, comprising 1.2 million cells across 66 coronal sections.

The human primary liver cancer (PLC) dataset includes five cases of hepatocellular carcinoma (HCC-1 to HCC-5), one case of intrahepatic cholangiocarcinoma (ICC-1) and one case of combined hepatocellular and cholangiocarcinoma (cHC-1), containing 84,823 spots in total. We selected one slice (HCC-1L, where L represents the leading-edge section) as the test set, and took the other 20 slices as the training set. Since the data volume of PLC by Visium is much less than the mouse brain datasets by MERFISH, we trained a GeST model with fewer layers and heads (4 transformer layers and 4 heads per layer). The slice for evaluation, HCC-1L, measured the spatial gene expression from tumor to normal tissue of one patient. We cropped an area of 100 spots containing the edge of the tumor as unseen spots (labeled as "Test"), and took all the other spots as seen spots (labeled as "Ref") (Figure A.3). After pretraining on 20 slices, we applied GeST to generate gene expression at the location of unseen spots based on the information of the rest seen spots. We visualized the meta-cell on the UMAP and results showed that it can well preserve the original data space (Figure A.9).

The Stereo-seq dataset has one sagittal section from the mouse brain, with in total of 60,000 data spots. Each spot is bin50 (25 μm), a typical resolution used for analyzing stereo-seq data. We segmented the cortex region from the right top corner as the test data, and used the rest of the tissue as the training data. We train GeST with the default model size.

The mouse brain ischemic dataset is collected from the ischemic hemisphere of mice subjected to photothrombosis (experiment group) and the ipsilateral hemisphere of sham mice (control group), and is sequenced by 10X Visium spatial transcriptomics platform. Each group has four coronal sections, containing 19,777 spatial transcriptomic spots in total. Han et al. (2024) annotated the spots with anatomical brain region labels, including the normal and ischemic regions. There are 425 DEGs in the ICA and 1263 in the PIA_P. Since GeST was pre-trained on only 1122 genes in the MERFISH dataset, we used the intersection of both dataset in the *in-silico* spatial perturbation experiment.

### A.2 ITERATIVE CELL GENERATION

Our model generated unseen cells based on the information from seen cells within a given window size. If the unseen region is larger than the given window size, an iterative generation is needed (as shown in Figure A.10). Given the spatial location of all unseen cells $X_{unseen} = \{s(x_{u1}), s(x_{u2}), ..., s(x_{uN})\}$, in each iterative round, we first find a subset of the seen cells, $X_s$ from all seen cells $X_{seen}$. Each cell in $X_s$ has at least one unseen cell in its available window. These cells will be used as the reference in this round. For each cell $x_s$ in $X_s$, we generate expression for unseen cells in its window. Relatively speaking, a subset of cells $X_{pre} = \{x_{p1}, x_{p2}, ..., x_{pM}\}$ from $X_{unseen}$ will have at least one gene expression estimation given from reference cells. For instance, we assume $x_{pi}$ has n estimations $\{g_{x1}(x_{pi}), g_{x2}(x_{pi}), ..., g_{xn}(x_{pi})\}$, where $g_{x.}$ represents a gene

vector function based on cell $x.$'s window. In practice, $g_{x.}(\cdot)$ is a vector where each dimension is the probability of a meta cell in the meta cell vocabulary. To get the final prediction for cell $x_{ui}$, We use the mean value of all these estimations as the final value and multiply it with the meta cell vocabulary:

$$g(x_{ui}) = \frac{1}{N} \sum_{j=1}^{N} g_{xj}(x_{ui}) \cdot \mathcal{C} \tag{11}$$

Then we update $X_{seen} = X_{seen} \cup X_{pre}$ and $X_{unseen} = X_{unseen} \setminus X_{pre}$ at the end of this round. We repeat the above process in each round until $X_{unseen}$ is empty.

### A.3 2D SINUSOID POSITIONAL ENCODING

We first unified the spatial coordinates to millimeters based on the CCF information. Since the absolute values of spatial coordinates in different slices are not comparable, for each training sequence, we converted the coordinates of each cell into relative coordinates. Specifically, the coordinate value of the central cell is subtracted from each cell. Then we anchored each coordinate into integers in a fixed range. In our experiments, we use [0,200) as default. Then we use 2D sinusoidal positional encodings to encode the two-dimensional coordinates into high dimension embeddings. Our approach is inspired by the method proposed in CellPLM, which employs sinusoidal functions to encode spatial coordinates in two dimensions. The encoding for a cell located at coordinates $(x, y)$ is formulated as:

$$\begin{aligned} \text{PE}_{(x,y),2i} &= \sin\left(\frac{x}{10000^{2i/d}}\right), & \text{PE}_{(x,y),2i+1} &= \cos\left(\frac{x}{10000^{2i/d}}\right) \\ \text{PE}_{(x,y),2j+d/2} &= \sin\left(\frac{y}{10000^{2j/d}}\right), & \text{PE}_{(x,y),2j+1+d/2} &= \cos\left(\frac{y}{10000^{2j/d}}\right) \end{aligned} \tag{12}$$

where $d$ is the total dimension of the positional encoding, and $i, j \in [0, d/4)$ specify the feature dimensions. This formulation extends the original sinusoidal positional encoding used in transformers to two dimensions, capturing both horizontal and vertical spatial variations.

### A.4 ALGORITHMS FOR CELL EXPRESSION QUANTIZATION

---

**Algorithm 1:** Construction of meta cell vocabulary.

---

**Data:** Spatial dataset $\mathcal{X} = \{x_1, x_2, x_3, \dots, x_n\}$; Number of meta cells $K$; Number of hierarchical labels at different levels $K_1, K_2, K_3$
**Result:** Meta cell vocabulary $\mathcal{C}$; Hierarchical labels of each meta cell $L_1, L_2, L_3$

$\mathcal{P} \in \mathbb{R}^{n \times p} \leftarrow$ Normalize gene expression $g(\mathcal{X})$ and perform PCA reduction to $p$ dimensions.
$\mathcal{P}_{\text{label}} \in \mathbb{R}^n \leftarrow$ Calculate labels of each cell in $\mathcal{P}$ by K-means algorithm with $k$ categories.
$\mathcal{C}_{\text{pca}} \in \mathbb{R}^{K \times p} \leftarrow$ Average $\mathcal{P}$ for each cluster label in PCA space.
$\mathcal{C}_{\text{expr}} \in \mathbb{R}^{K \times T} \leftarrow$ Average $\mathcal{P}$ for each cluster label in expression space.
$L_1, L_2, L_3 \in \mathbb{R}^K \leftarrow$ Calculate labels of $\mathcal{C}_{\text{pca}}$ by K-means with $K_1, K_2, K_3$ clustering numbers.

---

**Algorithm 2:** Query for meta cell vocabulary.

---

**Input:** Query spatial expression profile $\boldsymbol{y} \in \mathbb{R}^T$
**Output:** Meta cell $\boldsymbol{c} \in \mathbb{R}^T$; Hierarchical labels of the meta cell $l_1, l_2, l_3$

$\boldsymbol{p} \leftarrow$ Project $\boldsymbol{y}$ to PCA reduction space
$i \leftarrow$ Retrieve the nearest neighbor index of $\boldsymbol{p}$ in $\mathcal{C}_{\text{pca}}$
$\boldsymbol{c} \leftarrow \mathcal{C}_{\text{expr}}[i]$
$l_1, l_2, l_3 \leftarrow L_1[i], L_2[i], L_3[i]$

---

Table A.1: Performance comparison of methods on 10X Visium PLC and Stereo Brain datasets.

| | 10X Visium PLC | | | Stereo-seq Brain | | |
|---|---|---|---|---|---|---|
| | **Spearman** | **RMSE** | **RMSE Top50** | **Spearman** | **RMSE** | **RMSE Top50** |
| MLP | 0.491 | 1.347 | 1.008 | 0.314 | 1.403 | 1.327 |
| GP | 0.272 | 1.357 | 1.200 | 0.073 | 1.413 | 1.402 |
| Ours | **0.499** | **1.320** | **0.950** | **0.323** | **1.399** | **1.324** |

## A.5 DETAILS FOR IN-SILICO SPATIAL PERTURBATION

***In-silico* activation**: For genes that were highly expressed in ICA (e.g. *Spp1, Anxa2, Rbp1*), we used a gaussian kernel function

$$f_{\text{act}}(x, y) = a \exp\left(-\frac{(x - x_0)^2 + (y - y_0)^2}{2\sigma^2}\right)$$

to replace the original expression in the ROI, where $a = \max g(x), x \in \{x_1, \ldots, x_n\}$ was the maximum value of all genes, $(x_0, y_0)$ was the center of the perturbation and $\sigma$ was a hyper-parameter for controlling the rate of decay.

***In-silico* inhibition**: For genes that were lowly expressed in ICA (e.g. *Lamp5, Slc17a7, Tafa1*), we used a the similar gaussian kernel function

$$f_{\text{inh}}(x, y) = g(x, y) \left[1 - \exp\left(-\frac{(x - x_0)^2 + (y - y_0)^2}{2\sigma^2}\right)\right]$$

where $g(x, y)$ represented the original expression at position $(x, y)$, $(x_0, y_0)$ was the center of the perturbation.

## A.6 EXTENDED FIGURES & TABLES

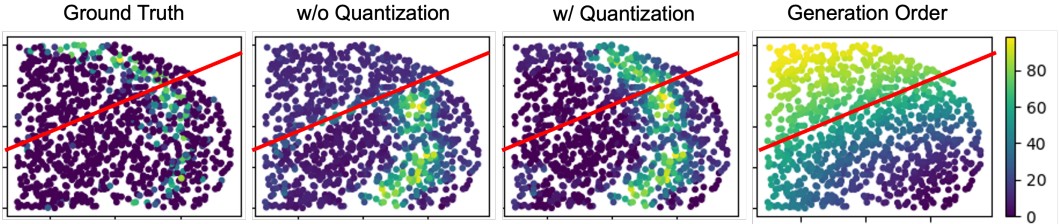

Figure A.1: Visualizing the effect of cell quantization on multiple steps generation. Regions below the red line are reference spots, and we generated all spots above the red line by following the order shown in the right sub-figure. The color in the first three figures represents the expression value of one marker gene.

Table A.2: AMI score of clustering results under different neighbor window sizes. The window size controls how many cells will be used for extracting niche embedding of the center cell. We report the mean ± standard deviation.

| **Level** | **Mode** | **200μm** | **400μm** | **600μm** |
|---|---|---|---|---|
| Division | Zeroshot | 0.416±0.164 | 0.448±0.180 | **0.470±0.174** |
| | Finetune | 0.452±0.166 | 0.495±0.173 | **0.501±0.173** |
| Region | Zeroshot | 0.449±0.102 | 0.473±0.106 | **0.484±0.107** |
| | Finetune | 0.490±0.116 | **0.517±0.127** | 0.515±0.077 |

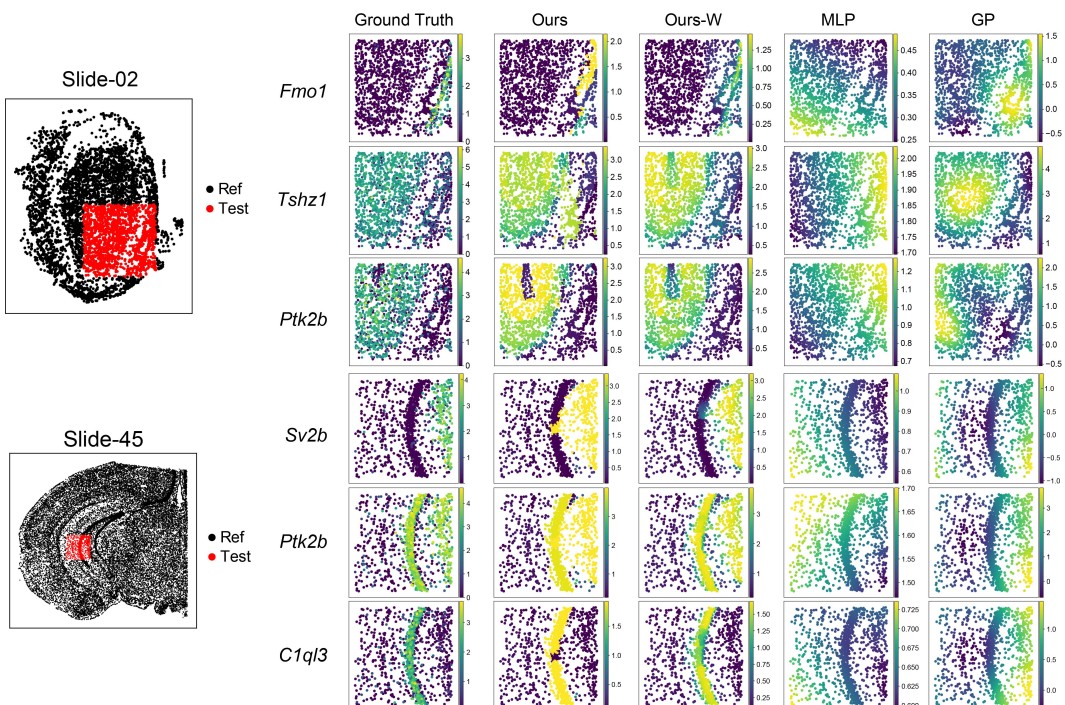

Figure A.2: Visualization of gene expression predictions. The black and red regions indicate the reference and unseen test regions, respectively. Each row on the right shows a gene's ground truth and predicted spatial patterns.

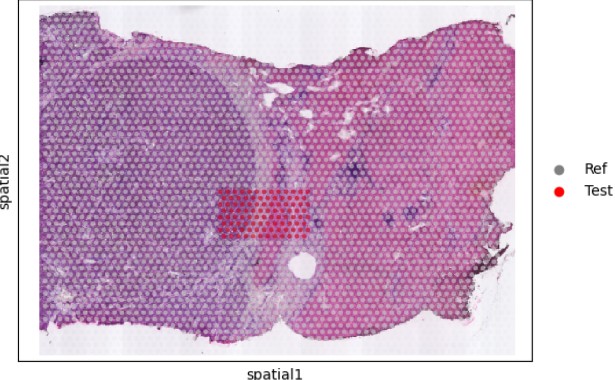

Figure A.3: Evaluation setting of sample HCC-1L from 10X Visium PLC dataset. Left half in deep purple is tumor tissue, and the rest right half is adjacent normal tissue. 100 spots containing the edge of tumor as are labeled as 'Test' set, and all the other spots are labeled as 'Ref' set.

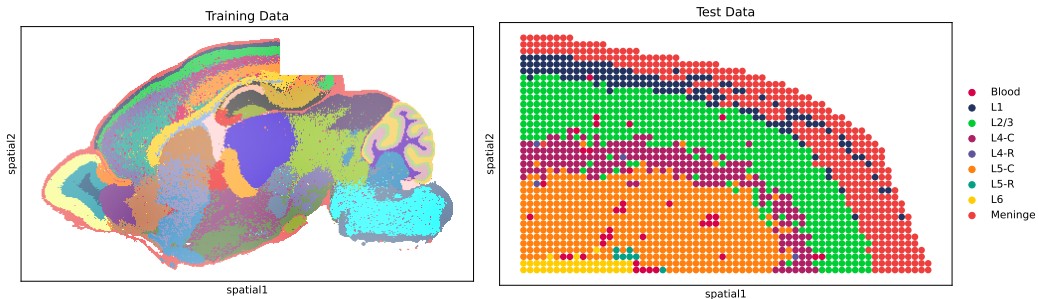

Figure A.4: Stereo-seq brain experiment setting. We used the fraction of the cortex layer as the test data and used the rest of the spots in the sagittal section as the training data.

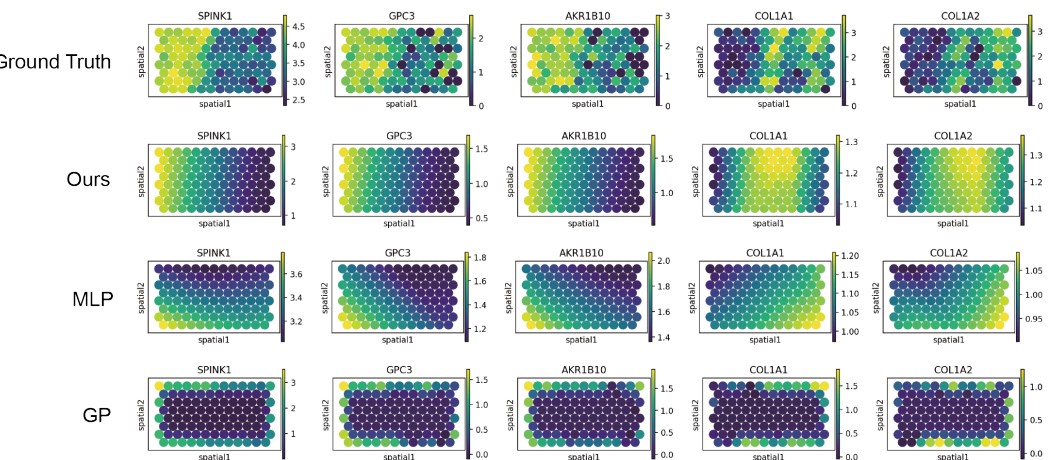

Figure A.5: Visualization of gene expression predictions of HCC-1L experiment.

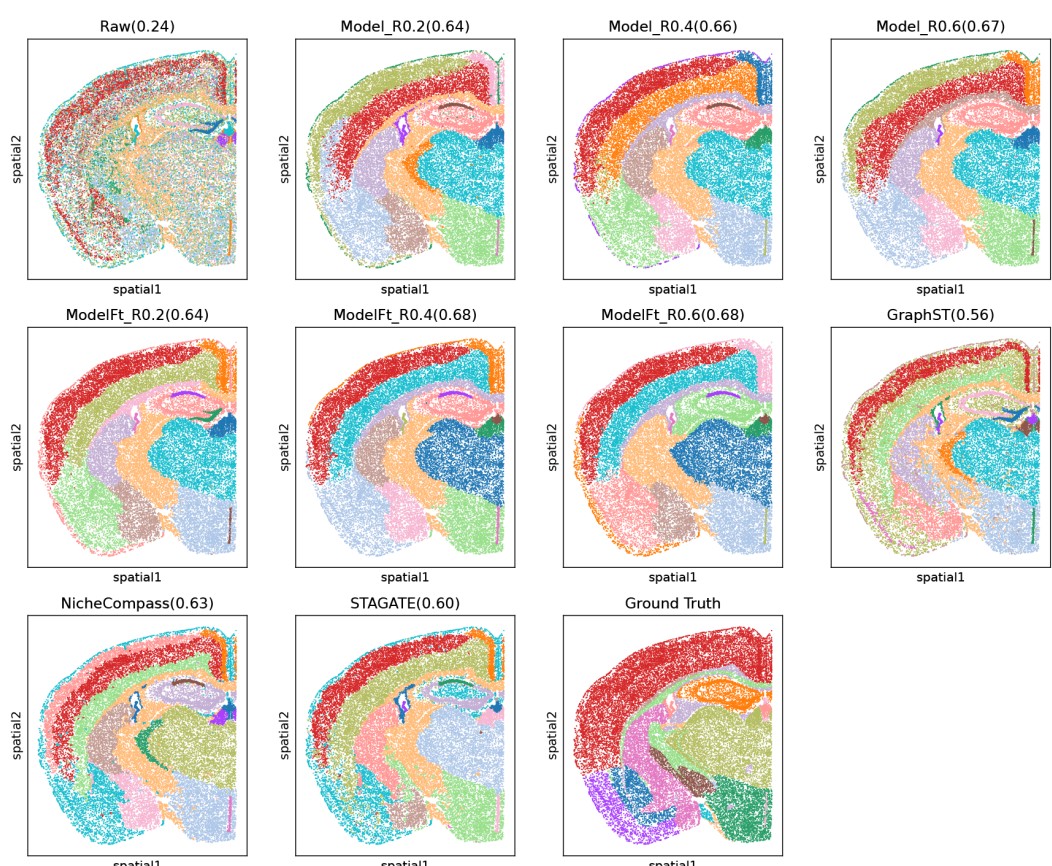

Figure A.6: Niche clustering results. Color represents clustered tissue regions at the division level. The number in the subtitle is the adjusted mutual information score.

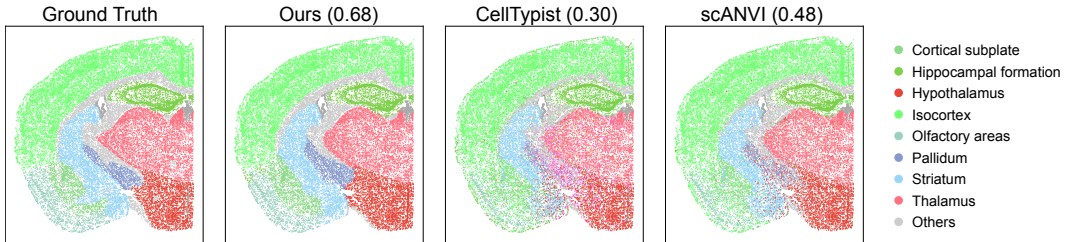

Figure A.7: Niche annotation results. Color represents annotated and predicted tissue divisions. The number in the subtitle is the macro F1 score.

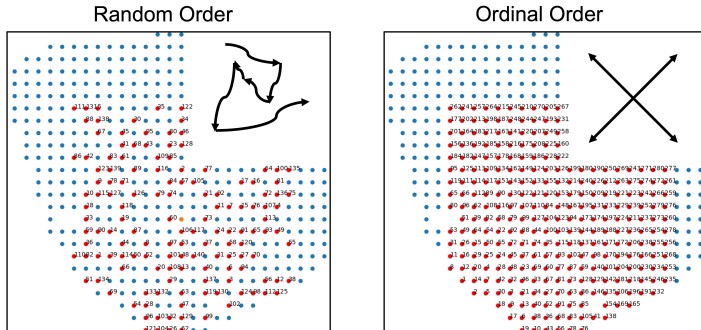

Figure A.8: Random order and ordinal order in spatial. Given all cells (in blue) in a section, the cells in red constitute a sequence for training. The number next to each cell represents the index in the sequence. In the random serialization strategy, numbers are scattered in space and the cells are not neared. In our proposed ordinal serialization strategy, numbers are sequentially assigned starting from the lower left (smallest x and y values) to the upper right (largest x and y values), but still retaining the randomness.

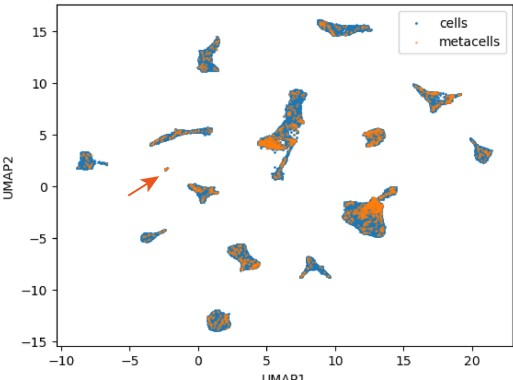

Figure A.9: UMAP plot of all meta cells and original cells from the PLC dataset. Orange arrow indicates one rare sub-cluster.

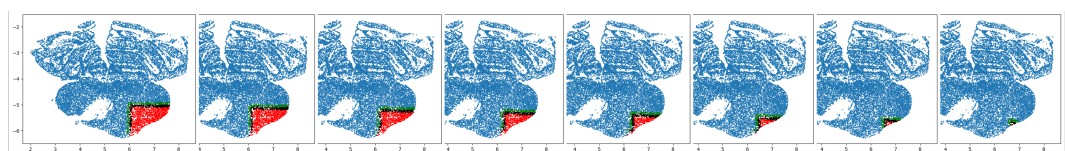

Figure A.10: Illustration of our iterative generation. Given seen cells (in blue) and spatial locations of unseen cells (in red), our model generated cells iteratively from left to right. For each round, it takes seen cells near the edge of the tissue as the refernce and generates adjacent unseen cells (in black), which will be used as the seen cells in the next generation.

Table A.3: Full list of differentially expressed genes (DEGs) in PIA_P. (n.s., not significant)

| | Gene name | DEG type (ground truth) | DEG type (prediction) |
|---|---|---|---|
| 1 | Col5a2 | High | High |
| 2 | Spp1 | High | High |
| 3 | Gfap | High | High |
| 4 | Ptprc | High | High |
| 5 | Lhfpl2 | High | High |
| 6 | Rnh1 | High | High |
| 7 | Ctss | High | High |
| 8 | Tmem176b | High | High |
| 9 | Dcn | High | High |
| 10 | Tgfbi | High | High |
| 11 | Prkcd | High | High |
| 12 | Cldn5 | High | High |
| 13 | Mdfic | High | High |
| 14 | Anxa2 | High | High |
| 15 | Fn1 | High | High |
| 16 | Tnc | High | High |
| 17 | Ucp2 | High | High |
| 18 | Maf | High | High |
| 19 | Cd44 | High | Low |
| 20 | Serpinf1 | High | High |
| 21 | Tmem176a | High | High |
| 22 | Cd24a | High | Low |
| 23 | Mcm6 | High | Low |
| 24 | Lsp1 | High | High |
| 25 | Serpina3n | High | n.s. |
| 26 | Col18a1 | High | Low |
| 27 | Lmo2 | High | High |
| 28 | Klk6 | High | High |
| 29 | Cd36 | High | n.s. |
| 30 | A2m | High | n.s. |
| 31 | Penk | High | n.s. |
| 32 | Cldn11 | High | High |
| 33 | Mafb | High | Low |
| 34 | Cdkn1a | High | Low |
| 35 | Lcp1 | High | Low |
| 36 | Fyb | High | n.s. |
| 37 | Lpl | High | Low |
| 38 | Rbp1 | High | High |
| 39 | Ctsc | High | Low |
| 40 | Tgfbr2 | High | High |
| 41 | Prdm8 | Low | High |
| 42 | Car4 | Low | High |
| 43 | Bhlhe22 | Low | High |
| 44 | Ccn3 | Low | High |
| 45 | Cnih3 | Low | n.s. |
| 46 | Krt12 | Low | n.s. |
| 47 | Slc30a3 | Low | n.s. |
| 48 | Pvalb | Low | n.s. |
| 49 | Chrm1 | Low | High |
| 50 | Fezf2 | Low | Low |
| 51 | Kcnj4 | Low | Low |
| 52 | Tafa1 | Low | Low |
| 53 | Coro6 | Low | Low |

| Continued on next page |
|---|

Table A.3 continued from previous page

| | Gene name | DEG type (ground truth) | DEG type (prediction) |
|---|---|---|---|
| 54 | Rgs6 | Low | Low |
| 55 | Neurod2 | Low | Low |
| 56 | Lamp5 | Low | High |
| 57 | Igfbp6 | Low | Low |
| 58 | Cpne9 | Low | Low |
| 59 | Pamr1 | Low | Low |
| 60 | Bcl11a | Low | Low |
| 61 | Adra1b | Low | Low |
| 62 | Dkkl1 | Low | n.s. |
| 63 | Cckbr | Low | Low |
| 64 | Chrm3 | Low | Low |
| 65 | Kcnh3 | Low | High |
| 66 | Slc17a7 | Low | Low |
| 67 | Bdnf | Low | Low |
| 68 | Myl4 | Low | Low |
| 69 | Epha4 | Low | Low |
| 70 | Cbln2 | Low | Low |
| 71 | Satb2 | Low | Low |
| 72 | Egr3 | Low | Low |
| 73 | Hs3st2 | Low | Low |
| 74 | Pde1a | Low | Low |
| 75 | Nwd2 | Low | Low |
| 76 | Mef2c | Low | Low |
| 77 | Rbp4 | Low | Low |
| 78 | Gabbr2 | Low | Low |
| 79 | Ldb2 | Low | Low |
| 80 | Neurod6 | Low | Low |
| 81 | Fgf13 | Low | Low |
| 82 | Kcnab3 | Low | Low |
| 83 | Sv2b | Low | Low |
| 84 | Satb1 | Low | Low |
| 85 | Adcy2 | Low | Low |
| 86 | Epha10 | Low | Low |
| 87 | Zmat4 | Low | Low |

Table A.4: Ablation study on spatial information

| | RMSE | RMSE50 | Spearman |
|---|---|---|---|
| Baseline | 1.367 | 1.214 | 0.29 |
| w/o Spatial | 1.397 | 1.325 | 0.23 |

