# OpenReview forum: "GeST: Towards Building A Generative Pretrained Transformer for Learning Cellular Spatial Context"
_ICLR.cc/2025/Conference — Submitted to ICLR 2025_

### Official Review · Reviewer_brAb · 2024-11-02

**Soundness:** 2
**Presentation:** 2
**Contribution:** 2
**Rating:** 3
**Confidence:** 4

**Summary:**

The authors introduce an innovative generative pre-trained model (GeST) designed to learn the spatial context of cells within spatial transcriptomics. This model ingeniously converts two-dimensional spatial data into a serialized one-dimensional sequence to accurately capture and model the intricate spatial relationships between cells. This novel approach facilitates a deeper understanding of complex tissue organizations and provides a promising direction for further research in the field. Preliminary results demonstrate the model’s effectiveness in capturing relevant spatial patterns, although further validation is required to assess its performance across diverse datasets and potential limitations in handling varying spatial resolutions.

**Strengths:**

The authors creatively employ a generative pre-trained transformer for the first time to understand spatial transcriptomics at the single-cell level, introducing innovative methods to the field.

**Weaknesses:**

1. The introduction could be enhanced by comparing the proposed GeST model with other relevant models like GraphGT or SpaGCN, as CellPLM, which is mentioned, differs significantly in context and isn’t directly related to spatial transcriptomics.
2. The model appears to primarily apply the Vision Transformer architecture to spatial transcriptomics with minimal modifications, suggesting a lack of substantial innovation.
3. The resolution variance among different spatial transcriptomics technologies should be more thoroughly addressed, potentially by incorporating datasets from Stereo-seq, Slide-seq v2, STARmap, and 10x Visium to provide a broader validation of the model’s utility. However, it is important to note that the resolution of 10x Visium is based on spots rather than individual cells. Does the model still perform effectively under these conditions?
3. In the ablation studies detailed in Table 3, it is unclear whether changes in the number of layers and heads simultaneously affect the window size. Clarification on how these architectural modifications impact the model’s spatial resolution would be valuable.
4. Consider the possibility of conducting an ablation study where the neighborhood  information is removed, to assess its impact on the model’s performance and spatial understanding.

**Questions:**

1. It is advisable to try data from multiple resolutions, as different technologies offer varying levels of resolution. For example, Stereo-seq can achieve subcellular resolution, which may allow the algorithm to examine the impact of organelles on the structure.
2. When conducting biological experiments involving tissue sections, these sections are often assembled from multiple pieces. This assembly process can introduce inaccuracies that may impact the reliability of neighborhood information. How should this issue be addressed to ensure data integrity?
3. As the authors mentioned, the model is expected to perform well in predicting genes with high spatial variation. It would be beneficial to validate this conclusion using cancer datasets, which are characterized by high variability. Additionally, considering cancer datasets could be crucial for addressing key questions about the tumor microenvironment.

---

> ### Author Response · Authors · 2024-11-25
> **Thank you for your comments!**
>
> We thank the reviewer for the constructive comments. We have shown additional discussions and experiments to strengthen our work further. The point-by-point responses to the comments are as follows. If our response does not fully address your concerns, please post additional questions and we will be happy to have further discussions.
>
> **WA1**: Thank you for your suggestion. We have updated the introduction section by adding these graph neural network works. It is now read as: "Rich ST datasets enable us to learn cell-cell relationships in a data-driven manner. Previous studies such as GraphST(Long et al., 2023) and SpaGCN (Hu et al., 2021) often trained graph neural network to integrate spatial and gene expression information. These models were trained independently for each dataset, leaving the paradigms of pretraining or generative modeling unexplored. A recent study..."
>
> **WA2**: Thank you for your feedback. Our model introduces significant innovations beyond ViT tailored to the unique characteristics of single-cell spatial transcriptomics data:
> 1. Input Design: Unlike ViT, which operates on fixed-order image patches, our input sequence integrates both positional tokens and gene expression tokens. This enables our model to process irregular spatial data and capture the spatial and molecular context effectively.
> 2. Specialized Attention Mechanism: In contrast to ViT's BERT style masking, which restricts attention to preceding tokens, our attention mechanism is specifically designed to explicitly incorporate positional and expression information for high computationally efficient training. This allows our model to better capture the spatial relationships and expression patterns essential for modeling single-cell data in spatial contexts.
> These modifications are fundamental to enabling our model to address the challenges of spatial transcriptomics, going beyond the scope of standard ViT applications.
>
> **WA3**: Thank you for your insightful comment. We agree that evaluating the model across different spatial transcriptomics technologies is crucial for assessing its robustness and transferability. In addition to the MERFISH brain dataset which is at the single cell resolution, we have performed a 10X Visium human primary liver cancer (PLC) dataset (multi-cellular resolution) and another Stereo-seq brain dataset (sub-cellular resolution). The detailed experiment design can be found in answers **Q1** and **Q3**. The results show that GeST continues to achieve strong generative performance, demonstrating its adaptability to diverse spatial resolutions and data sources. These findings have been included in the manuscript for broader validation.
> **Table**: Performance comparison of methods on 10X Visium PLC and Stereo Brain datasets.
>
> |       Method       | **Spearman** (10X PLC) | **RMSE** (10X PLC) | **RMSE Top50** (10X PLC) | **Spearman** (Stereo Brain) | **RMSE** (Stereo Brain) | **RMSE Top50** (Stereo Brain) |
> |--------------------|------------------------|--------------------|--------------------------|-----------------------------|-------------------------|-------------------------------|
> | **MLP**           | 0.491                  | 1.347              | 1.008                   | 0.314                       | 1.403                   | 1.327                        |
> | **GP**            | 0.272                  | 1.357              | 1.200                   | 0.073                       | 1.413                   | 1.402                        |
> | **Ours**          | **0.499**              | **1.320**          | **0.950**               | **0.323**                   | **1.399**               | **1.324**                    |
>
> **WA4**: Thank you for raising this important point. We appreciate your suggestion to further analyze the interplay between the number of layers, heads, and window size. We would like to clarify based on the results presented in Table 4.  First, with the number of layers fixed, we varied the window size. Performance improved as the window size increased from 200µm to 600µm, peaking at 600µm. This suggests that window size determines input information density, critical for achieving optimal performance. However, increasing the window size to 800µm reduced performance, indicating that an excessively large window size can be detrimental. We further examined the effect of increasing the number of layers with an 800µm window size. This configuration outperformed others, showing that larger window sizes require more layers to process the additional information effectively. These results highlight that underfitting, rather than overfitting, is a greater risk, and sufficient layers are essential for optimal performance.

---

> > ### Author Response · Authors · 2024-11-25
> > **Thank you for your comments! (continue)**
> >
> > **WA5**: Thank you for your suggestion. We have conducted a new ablation study to evaluate the impact of neighborhood information on the model’s performance. In this experiment, we replaced all spatial positional embedding with all-one vector, which means no neighbor information can be used for generation. The results show a significant drop in performance, highlighting its critical role in enhancing the model’s spatial understanding. We have added this result to the updated manuscript as Table A4.
> >
> > **Table**: Ablation study on spatial information.
> >
> > |            Method            | **RMSE** | **RMSE50** | **Spearman** |
> > |------------------------------|----------|------------|--------------|
> > | **Baseline**                | 1.367    | 1.214      | 0.29         |
> > | **w/o Spatial**             | 1.397    | 1.325      | 0.23         |
> >
> > **Q1**: We have added a new Stereo-seq experiment to show our model's performance.  In this dataset, we get the 'bin50' level dataset on a sagittal brain section, which corresponds to 25μm resolution and is the same resolution used in the original study. We split the whole tissue into a training and test region specified on cortex layers, and we also trained the MLP and gaussian process model as the baseline. Compared with these two methods, our model still achieves the highest performance. We have added these results to the updated manuscript.
> >
> > **Q2**: Thank you for your question. In our current approach, we mitigate random noise in spatial coordinates by anchoring all input coordinates into a spatial grid, which shares a similar idea from CellPLM and helps maintain robustness against small randomness around the anchor. However, for systematic errors introduced during the experimental assembly process, such as deformation or distortion of the tissues, we have not implemented specific corrections within our model. We believe such issues are best addressed during the preprocessing stage, where systematic biases can be identified and corrected to ensure data integrity before downstream analysis. This is an important consideration for enhancing the reliability of neighborhood information, and we will include this discussion in future work.
> >
> > **Q3**: Thanks for your valuable suggestion. According to your advice, we have further included an experiment on a cancer dataset. Wu et al. (2021) have performed 10X Visium spatial transcriptomics on human primary liver cancer (PLC) from 21 tissue specimens, including five cases of hepatocellular carcinoma (HCC-1 to HCC-5), one case of intrahepatic cholangiocarcinoma (ICC-1) and one case of combined hepatocellular and cholangiocarcinoma (cHC-1), containing 84,823 spots in total. We selected one slice (HCC-1L, where L represents the leading-edge section) as the test set, and took the other 20 slices as the training set. Since the data volume of PLC by Visium is much less than the mouse brain datasets by MERFISH, we trained a GeST model with 4 transformer layers and 4 heads per layer.
> > The slice for evaluation, HCC-1L, measured the spatial gene expression from tumor to normal tissue of one patient (Figure A3 in the updated manuscript). We cropped an area of 100 spots containing the edge of tumor as unseen spots (labeled as 'Test'), and took all the other spots as seen spots (labeled as 'Ref'). After pretraining on 20 slices, we applied GeST to generate gene expression at the location of 'Test' spots based on the information of the rest 'Ref' spots in HCC-1L. In a comparison of the two baseline models, our model achieves the highest Spearman coefficients as well as the lowest RMSE of all genes and the top 50 spatially variable genes (SVGs) (Refer to the table in WA3). Specifically, marker genes of malignant cells (SPINK1, GPC3, AKR1B10) and fibroblasts (COL1A1, COL1A2) are predicted to have clear zones, which are consistent with the ground truth. By contrast, the two baseline models failed to depict these spatial patterns (Figure A5 in the updated manuscript).
> > It is reported in the research by Wu et al. (2021) that PLC is characterized by high variability in spatial structure and cellular profiles. Under such a challenging scenario, our model can still capture and predict meaningful spatial patterns at the edge of the tumor, demonstrating the ability to handle cancer datasets and learn the characteristics of the tumor microenvironment. We have added these results to the revised manuscript.

---

### Official Review · Reviewer_2o83 · 2024-11-03

**Soundness:** 2
**Presentation:** 3
**Contribution:** 2
**Rating:** 5
**Confidence:** 4

**Summary:**

I'm initially rating this paper as "5: marginally below the acceptance threshold".

Summary: the paper proposes an auto-regressive generative model for spatial transcriptomic data. A notion of "order" is introduced thereby making use of (modified version of) pipelines for sequences with incremental updates. The method is evaluate on niche clustering, niche label annotation, unseen cell generation, and spatial perturbation prediction. To facilitate the generation of the final counts, a hierarchical clustering and meta-cell vocabulary is used.

**Strengths:**

- Clear writing and explanatory figures
-

**Weaknesses:**

- typo (not included in score): In line 236 the sentence shouldn't spot at "$g(x)$. Instead ..."

**Questions:**

- My main question/concern is that there is no inherent order in cells located in spatial positions (as mentioned in the paper). Lines 150-160 explain a procedure to assign a "pseudo-order" to cells. This procedure contains cropping a square from the spatial data, selecting one of the anchors, and repeatedly selecting cells based on their spatial distance to the selected anchor. At least I do not intuitively understand why such a procedure should resemble "an order"?
- For evaluating the generative power of the model, in Figure. 4 metrics like RMSE and correlation are used. Was there a reason for not using the commonly used metrics for this purpose, like Wasserstein distance, MMD, EMD, etc?

---

> ### Author Response · Authors · 2024-11-25
> **Thank you for your comments!**
>
> Thank you for your valuable feedback. Here we provide point-by-point responses to clarify our model and experiment design. If our response does not fully address your concerns, please post additional questions and we will be happy to have further discussions.
>
> **WA1**: Thank you for pointing out the typo in line 236. We have corrected the sentence.
>
> **Q1**: Thank you for raising this important concern. Transformer models, including ours, require sequential inputs for auto-regressive generation. In two-dimensional data modeling, such as images and spatial transcriptomics (ST), this necessitates converting spatially unordered data into a sequence. For example, in image processing, methods like Vision Transformer (ViT) are designed for regular grid-like image data. However, they are not suitable for handling the irregular spatial structures typical of ST data. To address this, we propose this pseudo-ordering strategy that serializes cells based on their spatial proximity and neighborhood gene expression, enabling our model to generate any cell's expression profile based on its neighborhood context.
>
> **Q2**: Thank you for your question. Our generated cells are based on specific spatial locations, and during evaluation, we directly match these generated cells to their ground-truth counterparts at the same coordinates. Since the ground truth for each location corresponds to a single cell rather than a distribution, it is not appropriate to use metrics like Wasserstein Distance (WD), Maximum Mean Discrepancy (MMD), or Earth Mover's Distance (EMD), which are designed for comparing distributions. Instead, we calculate paired metrics like RMSE and correlation by averaging the generated results and comparing them directly to the ground truth.

---

### Official Review · Reviewer_B5ci · 2024-11-04

**Soundness:** 3
**Presentation:** 2
**Contribution:** 2
**Rating:** 5
**Confidence:** 5

**Summary:**

The authors present a generative pre-trained transformer model designed for spatial transcriptomics. The authors propose strategies to tackle common challenges in applying transformer models to ST data, including a serialization strategy, cell quantization method, and spatial attention mechanism.

**Strengths:**

- The authors adopted a clever strategy to tokenize continuous gene expression profiles into discrete cell states. In particular, this helps mitigate error accumulation in autoregressive generation, a common issue when dealing with continuous data in transformer models.

- The model demonstrates strong performance across multiple tasks, including unseen cell generation, niche clustering/annotation, and in-silico spatial perturbation analysis. This versatility showcases the model's potential as a foundation for various spatial transcriptomics applications.

**Weaknesses:**

- The cell quantization strategy presented in the paper is not significantly different from previous strategies employed by existing methods for ST. For example, this problem of discretizing spatial data is addressed before by [1] Wen et al., [2] Yarlagadda et al, [3] Schaar et al.

- The evaluation is focused mainly on mouse brain datasets - which are known to have organized spatial structures of various distinct cell types. Evaluating model on more challenging datasets like from cancerous tissues will help solidify the work.  While spatial serialization introduces an ordinal structure, its application might overlook the full potential of irregular spatial patterns within tissues, limiting the model’s adaptability across different spatial configurations.

- The multi-level cell quantization and hierarchical loss approach are suited for well preserved mouse brain tissues. But in practice, ST data have several artifacts due to poorly preserved tissues and not very clean - transformer models for ST tend to perform relatively poorly compared to their CNN counterparts for modeling hierarchical information in the tissues.

- The reliance on a vocabulary to tokenize gene expression may lead to loss of subtle gene-level variations, potentially limiting the granularity of predictions, especially for rare cell subtypes.

- The model’s design does not fully account for dynamic gene-gene interactions within perturbed cells during in-silico simulations, which could lead to oversimplified, and often incorrect, biological interpretations.

-  The Spatial Attention mechanism is computationally expensive, and not optimized for long-range dependencies in large tissue sections, which may lead to biased local predictions without sufficient contextual global information. The pre-training is computationally intense and require multiple GPUs - and the authors should report how the model generalizes to non-brain tissues without sufficient available ST data.

- Authors use RMSE and Spearman correlation for evaluation, but lacks biologically relevant validation metrics, such as alignment with known cell types or tissue architectures.

- While the authors mention error accumulation in autoregressive generation, they don't provide a detailed analysis of how this affects long-range predictions or the model's stability over multiple generation steps.

[1] Wen, Hongzhi, et al. "Single cells are spatial tokens: Transformers for spatial transcriptomic data imputation." arXiv preprint arXiv:2302.03038 (2023).

[2] Yarlagadda, Dig Vijay Kumar, Joan Massagué, and Christina Leslie. "Discrete representation learning for modeling imaging-based spatial transcriptomics data." Proceedings of the IEEE/CVF International Conference on Computer Vision. 2023.

[3]. Schaar et al. "Nicheformer: a foundation model for single-cell and spatial omics." bioRxiv (2024): 2024-04.

**Questions:**

- What is the impact of tissue preparation methods and batch effects on GEST's performance?

How does the model,
- handle rare cell types or spatially isolated cells that may not have sufficient neighboring context?
- perform across different tissue types beyond the mouse brain?
- compare to graph-based approaches for ST data, such as spaGCN?

---

> ### Author Response · Authors · 2024-11-25
> **Thank you for your comments!**
>
> We thank the reviewer for the constructive comments. We have shown additional discussions and experiments to strengthen our work further. The point-by-point responses to the comments are as follows. If our response does not fully address your concerns, please post additional questions and we will be happy to have further discussions.
>
> **WA1**: We thank you for your comment on this important point and we would like to clarify that our approach is substantially different from existing methods and has not been employed in previous work. By detailing checking the provided paper, the first-mentioned work only uses raw continuous expressions, and the second and third-mentioned works applied VQ-VAE or the ranking method to quantize genes, where each quantized token corresponds to an image patch or a gene. However, our method does not discretize genes but cells, and thus each of our meta-cells tokens represents a cell profile.  By introducing meta cells, we could enhance the stability and efficiency of Transformer training. Besides, this discrete tokenization reduces potential errors in autoregressive generation and allows the model to better capture complex spatial patterns.
>
> **WA2**: Thank you for your valuable feedback. We agree that evaluating our model on more challenging datasets is essential to demonstrate its robustness and adaptability. Following your suggestion, we have further conducted experiments on spatial transcriptomics data of human primary liver cancer (PLC), which presents irregular spatial patterns and complex cellular heterogeneity.
> Wu et al. (2021) have performed 10X Visium spatial transcriptomics on PLC from 21 tissue specimens, including five cases of hepatocellular carcinoma (HCC-1 to HCC-5), one case of intrahepatic cholangiocarcinoma (ICC-1) and one case of combined hepatocellular and cholangiocarcinoma (cHC-1), containing 84,823 spots in total. We selected one slice (HCC-1L, where L represents the leading-edge section) as the test set, and took the other 20 slices as the training set. Since the data volume of PLC by Visium is much less than the mouse brain datasets by MERFISH, we trained a GeST model with 4 transformer layers and 4 heads per layer.
> The slice for evaluation, HCC-1L, measured the spatial gene expression from the tumor to the normal tissue of one patient (Figure A3 in the updated manuscript). We cropped an area of 100 spots containing the edge of the tumor as unseen spots (labeled as 'Test'), and took all the other spots as seen spots (labeled as 'Ref'). After pretraining on 20 slices, we applied GeST to generate gene expression at the location of 'Test' spots based on the information of the rest 'Ref' spots. In comparison to the two baseline models, our model achieves the highest Spearman coefficients and lowest RMSE.
>
> |       Method       | **Spearman** | **RMSE** | **RMSE Top50** |
> |--------------------|--------------|----------|----------------|
> | **MLP**           | 0.491        | 1.347    | 1.008          |
> | **GP**            | 0.272        | 1.357    | 1.200          |
> | **Ours**          | **0.499**    | **1.320**| **0.950**      |
>
> Specifically, marker genes of malignant cells (SPINK1, GPC3, AKR1B10) and fibroblasts (COL1A1, COL1A2) are predicted to have clear zones, which are consistent with the ground truth. By contrast, the two baseline models failed to depict these spatial patterns (Figure A5 in the updated manuscript).
>
> As reported in the research by Wu et al. (2021), PLC has various spatial architectures and complex cellular heterogeneity. Nevertheless, our model managed to capture and generate meaningful spatial patterns within these cancerous tissues, indicating the ability to handle diverse and irregular spatial configurations beyond organized structures like those in the mouse brain. We have updated the main manuscript and supplementary figures to include these new findings, which further strengthen the evaluation of our method.
>
> As for your concern about our serialization strategy, we acknowledge that such ordinally serializing an irregular spatial data structure may bring inductive bias to our modeling. To alleviate this potential bias, the final estimation of one cell's expression is taking the average of n predctions by sampling n sequences from its neighbor cells (Appendix A.2 in the manuscript). Experiment results on the PLC dataset also support the effectiveness of this design.

---

> ### Author Response · Authors · 2024-11-25
> **Thank you for your comments! (continue)**
>
> **WA4**: We acknowledge that quantizing continuous expression values to discrete cell states could lose subtle variations. This is a common trade-off between rich information with higher noise and less information with lower noise. In order to preserve a spectrum of cellular profiles as wide as possible, we chose a number of meta-cells large enough to ensure that the vocabulary could cover all cell types in the dataset. In other words, the vocabulary represents a much finer granularity than cell types or subtypes. For example, we built the meta-cell vocabulary with K=2000 for the PLC dataset. As shown in the UMAP plot (Figure A.9 in the updated manuscript), the meta cells broadly capture the distribution of all cells in the dataset, even including a rare cell sub-cluster. Therefore, we believe that cell quantization is a well-balanced strategy for auto-regressive modeling in spatial transcriptomics data.
>
> **WA5**: Thank you for your comment. We admit that fully modeling the dynamic gene-gene interaction within a cell requires a more sophisticated model. In our experiment, to alleviate this issue and guarantee the correctness of perturbation, we did the same activation or inhibition perturbation on multiple genes that were reported by previous studies to behave coherently.  And the results Fig. 5b showed our results achieve higher performance compared with baseline. We have changed the sentence "we adjusted expression values without considering gene-gene interactions within perturbed cells" into "our current design does not fully account for dynamic gene-gene interactions which similfies the biological mechanism" to highlight this point more in the conclusion section.
>
> **WA6**: Similar to approaches in natural language processing, GeST is pre-trained on shorter ranges to ensure computational efficiency. During fine-tuning, we can extend the model to handle longer sequences, enabling it to capture more global contextual information without significantly increasing computational costs. This strategy allows GeST to model long-range dependencies in large tissue sections effectively.
> To address the concern about generalization to non-brain tissues with limited available ST data, we have conducted additional experiments on human primary liver cancer (PLC) dataset with only 21 tissue slices. The results in WA2 demonstrate that GeST generalizes well and maintains robust performance in these challenging settings.
>
> **WA7**: In addition to RMSE and Spearman correlation, we have performed cell type classification and tissue architecture alignment tasks to validate our model's biological relevance in Section 4.2 and 4.3.  These evaluations were designed to demonstrate the biological relevance and applicability of our model.
>
> **WA8**: We thank you for your highlight of this point. We have added one supplementary figure (Figure A1) in the updated manuscript to visualize the difference between w/o quantization and w/ quantization models for a multiple-step generation. The model without quantization will lose the gene expression pattern.
>
> **Q1**: Thank you for your insightful question. In our current experimental setup, the models are trained on tissues prepared under the same laboratory conditions and using the same techniques. We do observe batch effects in the PLC dataset. Despite this, GeST demonstrates superior performance on unseen slides compared to other methods, underscoring the robustness of our cell quantization strategy. Addressing the integration of slides from diverse sources or preparation techniques is indeed an important direction for future work. Potential approaches include removing batch effects as a preprocessing step for all GeST models or explicitly incorporating batch correction during the cell quantization process. We appreciate your comment and have included this discussion in the updated version.
>
> **Q2**: Thank you for your thoughtful question. Our model addresses rare cell types and spatially isolated cells in two different cases:
> 1. Rare Cell Types: As replied in WA4, the meta cell vocabulary offers a much finer granularity than cell types or subtypes in the dataset by a large number of meta cells (K=2000). If users are concerned about the coverage of meta cells, they can increase K and validate the meta cell distribution by plotting meta-cells and all cells on the same UMAP.
> 2. Spatially Isolated Cells: For spatially isolated cells, increasing the window size can help model long-range spatial correlations. In our current setup, with a 600 µm window size, the maximum sequence length is ~800 cells. Users can extend this to capture broader spatial contexts. However, we acknowledge that for cells in highly vacant regions, spatial models may become less effective as these cells are minimally influenced by spatial context. In such cases, alternative generation strategies may be required.
> We appreciate your comment and would like to consider expanding on this discussion in future work.

---

> > ### Author Response · Authors · 2024-11-25
> > **Thank you for your comments! (continue)**
> >
> > **Q3**: We really appreciate your suggestion. We have added experiments on human primary liver cancer (PLC). Please check WA2 for details.
> >
> > **Q4**: Thank you for your suggestion. In the previous manuscript, we compared our methods with two graph methods (STAGATE and GraphST), and we have included performance results from SpaGCN on the niche clustering task in this revision.
> >
> > **Table**: AMI score of different methods for niche clustering results at both the region and division levels. We report the mean ± standard deviation. *NicheC*: NicheCompass, *Ours-Ft*: Ours fine-tuned model.
> > | **Level**   | **Ours**            | **GraphST**       | **NicheC.**       | **SpaGCN**        | **STAGATE**       | **Raw**           | **Ours-Ft**       |
> > |-------------|---------------------|-------------------|-------------------|-------------------|-------------------|-------------------|-------------------|
> > | **Division** | **0.469 ± 0.173**  | 0.388 ± 0.152     | 0.438 ± 0.177     | 0.201 ± 0.070     | 0.420 ± 0.167     | 0.183 ± 0.091     | **0.470 ± 0.174** |
> > | **Region**   | **0.484 ± 0.107**  | 0.414 ± 0.091     | 0.481 ± 0.113     | 0.230 ± 0.067     | 0.462 ± 0.114     | 0.244 ± 0.077     | **0.515 ± 0.077** |
> >
> > Our model outperforms both methods, demonstrating its effectiveness in this application. Besides, we have also added the relevance of SpaGCN and other graph-based approaches into the introduction section.
> >
> > It is now read as: "Rich ST datasets enable us to learn cell-cell relationships in a data-driven manner. Previous studies such as GraphST(Long et al., 2023) and SpaGCN (Hu et al., 2021) often trained graph neural network to integrate spatial and gene expression information. These models were trained independently for each dataset, leaving the paradigms of pretraining or generative modeling unexplored. A recent study..."

---

> > > ### Comment · Reviewer_B5ci · 2024-12-03
> > >
> > > I would like to thank the authors for their responses to the reviewers' comments. The concern about novelty of the work still hasn't been satisfactorily addressed. I will maintain my current rating.

---

### Author Response · Authors · 2024-11-29
**Response to Common Questions and Concerns**

We sincerely thank all reviewers for their thoughtful and constructive feedback. Guided by these suggestions, we have made significant revisions and introduced new experiments. These new findings further solidify our novel and effective design for generative modeling of spatial transcriptomic data, support the generalizability of our model, and also expand the scope of our study. Below, we summarize the key updates:
1. **Extended Evaluation of Model Generalizability to Cancer Dataset**: In response to the reviewers' concern on model generalizability to other tissues, we have extended our model applications to the 10X Visium human primary liver cancer (PLC) dataset (Figures A3). Despite the challenges of irregular spatial architecture and complex cellular heterogeneity, our model outperformed other baselines in terms of generative ability (Table A1). Notably, in the tumor leading-edge region, GeST was the only method capable of recovering unseen spatial patterns of marker genes for malignant cells and fibroblasts (Figures A5).
2. **Broadened Applications of More Technologies with Various Resolutions**: Another new experiment on Stereo-seq mouse sagittal brain dataset also indicated the superior generative performance of our model (Figures A4, Table A1). Taken together, we have validated GeST's broad learning ability for Visium (multi-cell resolution), MERFISH (single-cell resolution) and Stereo-seq (sub-cell resolution), which are detailed in Section 4.1, Appendix A.1.
3. **Expanded Ablation Studies**: We have added more ablation studies to show the effectiveness of different modules.
    - We trained a version of our model without spatial neighborhood information, which resulted in significantly reduced performance (Table A4).
    - We clarified that our cell quantization approach is an unsupervised method independent of cell-type labels. By building the meta cell vocabulary of a large size, we demonstrated the broad coverage of cell distribution in the dataset (Figure A9). Removing this strategy would cause model failure (Figure A1), highlighting its important role in mitigating error accumulation.
4. **Comparison with Additional Methods:** We conducted a niche clustering experiment using SpaGCN and expanded the comparison to include five additional methods, spanning both graph- and transformer-based approaches (Table 1).

We have carefully revised the manuscript to incorporate these updates. We also appreciate the reviewers’ interest on the GeST model’s strengths in spatial cell generation and in-silico spatial perturbation. We look forward to addressing any additional comments or feedback.

---

### Meta-Review · Area_Chair_UV5n · 2024-12-20

**Metareview:**

Reviewers highlighted issues with the model's novelty, as its design heavily resembles existing architectures with minimal innovations tailored to spatial transcriptomics. Furthermore, evaluations were limited to specific datasets, with insufficient exploration of more complex spatial patterns or diverse tissue types. While the authors' rebuttal addressed some concerns through additional experiments and clarifications, key questions regarding scalability, computational efficiency, and biological validation remained inadequately resolved.

**Additional Comments On Reviewer Discussion:**

During the reviewer discussion, key concerns were raised about the novelty of the proposed GeST model, its applicability to diverse datasets, and the adequacy of its biological validation. The authors responded with additional experiments, including evaluations on cancer datasets and ablation studies, and clarified their model's architectural contributions. However, reviewers remained unconvinced about the model’s originality, scalability, and computational efficiency, and noted that critical concerns about dynamic gene-gene interactions and validation metrics were insufficiently addressed. While the additional experiments strengthened the submission in certain areas, the unresolved core concerns ultimately weighed heavily in the decision to reject the paper.

---

### Decision · Program_Chairs · 2025-01-22

Reject